# Importance of the Host Phenotype on the Preservation of the Genetic Diversity in Codling Moth Granulovirus

**DOI:** 10.3390/v11070621

**Published:** 2019-07-05

**Authors:** Benoit Graillot, Christine Blachere-López, Samantha Besse, Myriam Siegwart, Miguel López-Ferber

**Affiliations:** 1LGEI, Ecole des Mines d’Alès, Institut Mines-Telecom et Université de Montpellier Sud de France, 6 Avenue de Clavières, 30319 Alès, France; 2Natural Plant Protection, Arysta LifeScience group, Avenue Léon Blum, 64000 Pau, France; 3INRA, 6, Avenue de Clavières, 30319 Alès, France; 4INRA, unité PSH, Agroparc, CEDEX 9, 84914 Avignon, France

**Keywords:** *Cydia pomonella* granulovirus, codling moth, biological control, genetic diversity, coevolution, selection pressure

## Abstract

To test the importance of the host genotype in maintaining virus genetic diversity, five experimental populations were constructed by mixing two *Cydia pomonella* granulovirus isolates, the Mexican isolate CpGV-M and the CpGV-R5, in ratios of 99% M + 1% R, 95% M + 5% R, 90% M + 10% R, 50% M + 50% R, and 10% M + 90% R. CpGV-M and CpGV-R5 differ in their ability to replicate in codling moth larvae carrying the type I resistance. This ability is associated with a genetic marker located in the virus *pe38* gene. Six successive cycles of replication were carried out with each virus population on a fully-permissive codling moth colony (CpNPP), as well as on a host colony (R_GV_) that carries the type I resistance, and thus blocks CpGV-M replication. The infectivity of offspring viruses was tested on both hosts. Replication on the CpNPP leads to virus lineages preserving the *pe38* markers characteristic of both isolates, while replication on the R_GV_ colony drastically reduces the frequency of the CpGV-M *pe38* marker. Virus progeny obtained after replication on CpNPP show consistently higher pathogenicity than that of progeny viruses obtained by replication on R_GV_, independently of the host used for testing.

## 1. Introduction

Codling moth, the main insect pest for apple and pear production [1], is widely distributed around the world. Most apple production areas suffer damages caused by this insect [2]. The repeated use of chemical insecticides led to the progressive development of resistance to most of them [3]. To sustain apple production, the necessity of alternative methods becomes evident.

Baculoviruses are authorized as biological control agents in field conditions because they are specific to one or few insect species and harmless for beneficial insects [4].

The first *Cydia pomonella granulovirus* (CpGV) isolate originates from Mexico (CpGV-M) [5]. This isolate is one of the most widely-used GV for biological control; it is considered as the reference isolate, and a representative clone of this isolate has been completely sequenced [6]. In Europe, all commercial formulations of CpGV before 2008 were derived from it [7].

Since 2004, resistance to CpGV-M has been reported in orchards, first in Germany [8] and France [9], later in other European countries [10]. The most common resistance is now called “type I resistance”, and it has been located at the Z chromosome of the insect [11]. CpGV is the prototype of the genus formerly called “granulovirus”, which now are named “*Betabaculovirus*”, one of the four genera of the *Baculoviridae* family. In their life cycle, baculoviruses alternate between two types of particles, the budded virus (BV), a virus particle where a nucleocapsid that contains the genome acquires a membrane of cellular origin when budding out of the cell, and an occluded form, where virus particles are protected within a proteinaceous paracrystalline structure, the occlusion body (OB). BVs are responsible for within-host cell infections, while OBs are responsible for infection from one host to another, via the external environment. In the alpha-, gamma-, and delta-baculoviruses, usually called nucleopolyhedroviruses (NPV), OBs can contain many independent virions, whose genomes might not be identical [12]. In the beta-baculoviruses, OBs usually contain a single virion, rarely two or more, and each virion contains a single genome. [13]. These OBs, smaller in size, are called granulae. The BVs of all baculoviruses contain a single nucleocapsid, thus a single genome.

Alpha-baculoviruses (prototype species *Autographa californica multiple nucleopolyhedrovirus*, AcMNPV) are characterized by an important genetic diversity, both within a single isolate or between isolates. Analysis by restriction fragment length polymorphism (RFLP) allowed the characterization of nine genotypes from a population of *Spodoptera frugiperda multiple nucleopolyhedrovirus* SfMNPV [14]. It is indeed common to find various genotypes within a single larva. Twenty-four genotypic variants of a baculovirus have been isolated from a single diseased *Panolis flammea* larval cadaver [15]. Conversely, GV isolates appeared as highly homogeneous when analyzed by RFLP [16], and the differences between isolates were also very limited [17,18,19], leading to the conclusion that “there appears to be very little genotypic variation between virus isolates” [17].

After detection of the resistance, by screening among available CpGV isolates, it was possible to find some isolates able to bypass the resistance [20,21], confirming that phenotypic variability is present. This phenotypic variability is indeed the result of a previously-unobserved genotypic diversity.

CpGV genotypic diversity has been then analyzed in more detail. All the genotypes described so far belong to five groups, A–E [20,22]. CpGV-M and CpGV-R5 belong respectively to Groups A and E. In contrast to NPVs, genetic diversity within a larva, that is, mixed infection, has only been described in a single isolate, I68 [23]. Analysis of the I68 isolate, originating from a single larva revealed the presence of two genotypes, belonging to groups C and A.

A modification of the viral *pe38* gene, whose function remains unknown, has been associated with the ability of a virus to replicate in a type I-resistant host, like R_GV_ [22]. CpGV-M and CpGV-R5 differ by a small 24-bp duplication, present in CpGV-M, but absent in CpGV-R5. In type I-resistant hosts, CpGV-M replication is blocked at an early stage in all cells of the larva [24].

Genetic diversity is an important factor for coping with the differences between the genetic background of host populations, and the continuously-evolving host defense capabilities [25,26].

Coinfection with multiple genotypes in a single cell is a general rule for AcMNPV [27]. It has also been demonstrated in SfMNPV [12], and it is probably true for all alpha-baculoviruses. These genotypes can act together to invade the host larvae [14,28,29]. In SfNPV, it has been observed that individual genotypes are less pathogenic to the host than a virus population that is genotypically diverse [14].

In NPVs, multiple genotypes can be occluded in the same OB [28], so one OB can thus represent a sample of the population diversity, and consequently, a larva that eats a single OB can be infected by a variety of virus genotypes. In GVs however, genotypic diversity in a single larva would rely on the larva eating more than one OB. In a previous study, we have investigated the ability of mixes of CpGV isolates to control a resistant colony. We observe that mixtures of OBs are more efficient than expected on the control of the hosts’ larvae, be they susceptible or resistant [30]. For that approach to be effective, each larva should ingest at least two CpGV OBs carrying different genotypes. The question arising is how genotypically diverse isolates will behave in coinfection conditions in the host upon successive cycles of replication. Here, we follow mixed genotype virus lineages for six cycles of replication on both susceptible and resistant insects and evaluate their evolution both at the phenotypic (how their efficacies evolve on each host) and the genotypic levels (how virus genotypic markers are maintained).

## 2. Materials and Methods

### 2.1. Insects

Two *Cydia pomonella* colonies were reared in the laboratory. The CpNPP colony was fully susceptible to CpGV-M. It was derived from a field population from the Loire Valley in France and replicated in the laboratory since the early 1990s. The R_GV_ colony was resistant to CpGV-M. This resistant colony was derived from a natural resistant population (St-A) found in the region of Saint-Andiol (Bouches-du-Rhône, France), followed by selection for resistance to CpGV-M as previously described [21]. Briefly, larvae were reared on media containing 100 OB/µL of CpGV-M until pupation; adults that emerged were allowed to mate, the eggs collected, and the new generation of larvae submitted to the same process. Continuous selection for eight generations allowed a survival higher than 99% of individuals. The R_GV_ colony harbored the sex-linked resistance now called type I resistance [11]. Both colonies were reared as previously described [31].

### 2.2. Viruses

CpGV-M (laboratory stock 2020-s1) and CpGV-R5 (laboratory stock 2016-r16), the two virus isolates used in this work, have been described in previous studies [21,31]. The CpGV-M is the reference isolate that fully replicates on CpNPP (LC_50_ = 13.10 OB/µL (6.55–23.20)), but not on R_GV_ ((LC_50_ = 2.22 × 10^6^ OB/µL (1.19 × 10^6^–5.67 × 10^6^). The CpGV-R5 isolate is able to overcome the resistance of the R_GV_ colony (LC_50_ = 22.43 OB/µL (13.73–34.36)) and also replicates on CpNPP (LC_50_ = 6.76 OB/µL (2.6–13.37)) [32].

### 2.3. Viral Populations

Five experimental virus populations were constructed by mixing in various proportions OBs of the two isolates. The proportions of each isolate in the mixed virus populations were 99% CpGV-M + 1% CpGV-R5, 95% CpGV-M + 5% CpGV-R5, 90% CpGV-M + 10% CpGV-R5, 50% CpGV-M + 50% CpGV-R5, and 10% CpGV-M + 90% CpGV-R5. Pure CpGV-M and CpGV-R5 were used as reference virus populations. These OBs mixtures are referred to as Passage 0 OBs (P0) [30].

From each virus population, two independent lineages were followed by passaging them during six passages on susceptible (CpNPP) or resistant (R_GV_) insects (Pi_CpNPP_ and Pi_RGV_, i being the passage number) (Figure 1).

### 2.4. Successive Passages of the Different Viral Lineages

Forty-eight 3rd–4th instar (7 days old) susceptible (CpNPP) or resistant (R_GV_) larvae were individually deposited in 24-well plates. Each well contained 1 mL formaldehyde-free diet (Stonefly Heliothis Diet, Ward’s Science, Rochester, NY, USA) inoculated with 50 µL of each P0 viral suspension at 800 OBs/µL per well, that is a concentration of 40 OBs/µL of diet. At this concentration, mortality higher than 90% was expected. Plates were then incubated at 25 °C (±1 °C) with a 16:8-h (light/dark) photoperiod and a relative humidity of 60% (±10%). Four days later, all larvae presenting signs of viral infection were extracted from the rearing diet, stored at 25 °C for one more day, then frozen (−20 °C). OBs were then extracted and purified as previously described [21]. This suspension constituted the first amplification (P1) of each viral lineage. This protocol was used to produce the following passages (P2–P6).

### 2.5. Bioassays

Bioassays were performed as previously described [21]. To summarize, 96-well plates were filled with about 200 µL of a formaldehyde-free artificial diet (Stonefly Heliothis Diet, Ward’s Science, Rochester, NY, USA). Six microliters of OB suspension at the required concentrations (from 2–6250 OBs/µL) were deposited over the surface of each well. Non-infected control wells were included (6 µL of distilled water). One neonate larva was then laid on each well. The plates were sealed with Parafilm™ and incubated as described previously. After 7 days, plates were checked and the mortality scored. These data were subjected to probit analysis [33] performed with the POLO + software [34]. Each test was repeated at least three times, and the results pooled when they appeared consistent (no statistically-significant differences between tests). An average of 566 larvae were used for each modality (between 344 and 1343). The experimental plan is summarized in Figure 1.

### 2.6. Estimating the Relative Proportions of Each Genotype by PCR

Genomic DNA was extracted from a sample of the OBs produced for each virus lineage at each passage. These samples were used as a matrix for PCR using the specific primers for the *pe38* gene region previously published [30]: CpGV 19003R (5’ ccggctgcagCGAGTCGAGCACCACCATTA 3’) and CpGV18705F (5’ cgcgggatccACGGTGTGTCATTAGCCACC 3’); the numbers refer to the nucleotide positions in the NC_002816 CpGV-M sequence. These primers amplify fragments of differing size in the two genotypes (295 bp for CpGV-R5 and 315 bp for CpGV-M), allowing us to discriminate them.

The PCR fragments were separated in a 3% agarose gel (NOVAGEL GQT, Conda S.A., Torrejon de Ardoz, Madrid, Spain) in TBE buffer and visualized on a UV transilluminator after ethidium bromide staining. No attempt at quantification of the relative frequencies of each genotype was done.

## 3. Results

Bioassays of the 10 virus lineages (5 different frequencies × 2 hosts) have been performed both on CpNPP and R_GV_ on OBs at passages P1, P3, and P6. The bioassays of the original mixes (P0) both on susceptible and resistant larvae have been previously published [30]. They are used here as a reference.

### 3.1. Lineages Obtained on Susceptible Insects

From P0–P1_CpNPP_, there was a reduction of the pathogenicity in all virus populations for CpNPP larvae. From P1_CpNPP_–P6_CpNPP_, the general trend was a progressive increase of the pathogenicity in the five virus lineages, roughly reaching the efficiency of the pure genotype virus populations, CpGV-M or CpGV-R5, on this permissive host (Figure 2a). Within each passage, however, there was an important variability on the CL_50_s, which did not correlate with the original genotype composition.

When testing these lineages on R_GV_ resistant insects (Figure 2b), a similar pattern was observed. At P1_CpNPP_, the various lineages responded quite differently, and the LC_50_s varied between 36 and 176 OBs/µL. This response did not correlate with the original content of CpGV-R5 at P0. At P6_CpNPP_, all virus lineages showed a similar behavior. Their pathogenicity was comparable to that of CpGV-R5 on this host.

### 3.2. Lineages Obtained on Resistant Insects

The results of the bioassays performed with the P1_RGV_–P6_RGV_ are presented in Figure 3. When the virus lineages were passaged on resistant insects, thus submitted to strong selection against the Group A genotype, their pathogenicity did not improve regularly over cycles, independent of the host colony used on the bioassay (LC_50_ varied from 20–80 OB/µL on CpNPP and from 40–190 OB/µL on R_GV_). In addition, the efficacy on both hosts was not correlated with the original frequency of the parental virus genotypes: M95-R05 P6_RGV_ showed the lowest LC_50_ on both R_GV_ and CpNPP, while M99-R01 P6_RGV_ had the highest LC_50_ on CpNPP, but not on R_GV_.

### 3.3. Genomic Composition of the Virus Populations

The diversity in the virus populations was examined by PCR using the differences at the *pe38* gene. The analysis of virus lineages after six passages on the susceptible (P6_CpNPP_) and the resistant host (P6_RGV_) revealed the retaining of genetic diversity in all populations of P6_CpNPP_, but such diversity was clearly reduced or absent in P6_RGV_ (Figure 4).

## 4. Discussion

Resistance to selection factors (insecticides, antibiotics) often implies an evolutionary tradeoff. In the presence of the selection factor, resistant individuals have a selective advantage compared to susceptible ones. Conversely, in the absence of the selection factor, susceptible individuals are favored due to their higher fitness. Under this hypothesis, when both genotypes are present in a population, a progressive increase of susceptible genotypes should be observed in the absence of the selection trait [35]. Two approaches can be taken to analyze tradeoffs, measuring fitness parameters (i.e., fecundity, viability, developmental time, size of offspring) or analyzing the changes in the frequencies of experimental populations composed of a mixture of genotypes.

Undorf-Spahn et al. [36] studied both some representative fitness parameters in the laboratory and the temporal evolution of insect populations. No apparent cost was observed for resistance to the CpGV-M by codling moth, suggesting that this resistance would persist in the orchard insect populations even in the absence of CpGV-M treatments.

From the virus perspective, no major differences were observed between CpGV-M and CpGV-R5 in the parameters analyzed (virus production, pathogenicity to susceptible insects). The two virus isolates replicated in the permissive host (CpNPP) at approximately the same level, in terms of OB production, and showed a slight difference in their LC_50_s (13.10 OB/µL compared to 6.76 OB/µL [31]). To complete this analysis, the temporal evolution of experimental populations constructed by mixing pure genotypes was set up.

By mixing OBs from the two virus isolates (P0), experimental virus populations were constructed covering all the range of proportions for each virus isolate. These experimental virus populations were then allowed to evolve on the two hosts for six successive passages, constituting 10 different lineages, five on each host. Their pathogenicity was evaluated on both hosts at P0, P1, P3, and P6.

As CpGV OBs contain one genome (exceptionally more than one), we have chosen conditions in which each larvae would eat more than one OB, thus, increasing the probability of the presence of multiple genotypes in the larvae. In the four-day infection, period a L3 larva ate approximately 500 µL of diet; at 40 OB/µL, that means in the range of 10^4^ OBs. However, the probability of successful infection with a second virus might be related to the delay between the first infection and the second one, as has been shown for SfMNPV [37].

For a given number of OBs ingested, the probability of multiple genotype infection depends on the relative frequency of the genotypes. It is maximal for the even distribution of the genotypes (50% of each in our experiments with only two genotypes) and decreases when increasing the differences in the relative frequencies. Accordingly, in the experimental virus population containing 99% CpGV-M + 1% CpGV-R5, the probability of having larvae infected by only one genotype will be higher than for the 50% CpGV-M + 50% CpGV-R5 population, for the same number of OBs ingested.

We hypothesized that the permissive CpNPP host would not influence the outcome of viral infections, while in the resistant R_GV_ host, the presence of CpGV-M genomes in the offspring will rely on the replication of at least one genome of CpGV-R5, as the LC_50_ for pure CpGV-M is far above the dose use for the test. It would be expected that if both viruses were almost equally fit in the permissive host, the virus lineages would maintain their genotypic diversity, while lineages replicating in the non-permissive host would eliminate the CpGV-M genotype.

We have previously demonstrated [30] that the two genotypes do not act independently of the infected larvae. The pathogenicity of the P0 mixed populations was greater than expected under the independent action hypothesis. In addition, CpGV-M replicated on R_GV_ when in co-infection with CpGV-R5; thus, P1_RGV_ contained CpGV-M genomes.

Surprisingly, there was an overall reduction in the efficacy against CpNPP between P0_CpNPP_ and P1_CpNPP_ (from an average LC_50_ of 15.66 OB/µL–67.52 OB/µL, respectively). This decrease was statistically significant for all but the M99 + 1R P1_CpNPP_, and this can be explained by the higher frequency of infections with only one genotype under this condition.

In plant or animal selection, it is common to observe an increase in the fitness of hybrids and a decrease when reproducing these hybrids on successive generations. The results we obtained resemble this effect, if we consider the whole inoculum ingested as the selection unit. The reduction of the efficacy of the control at P1_x_ could be the consequence of diversity generation via recombination, leading to some less adapted gene combinations that would be counter selected in the following passages. Recombination of NPV has been demonstrated in cell cultures [38] and in vivo, but co-infection of host cells by multiple GV genotypes has never been analyzed. Our results could partially be explained by such a mechanism, but this hypothesis implies that the resistance breaking is not exclusively due to the modification of the *pe38* gene. Further work will be required to explore this point.

In the successive passages, this decrease in efficacy was compensated; at P6_CpNPP_, the LC_50_s of all lineages recovered to the level of the parent viruses on CpNPP. When tested on the R_GV_, the host in which these lineages did not replicate previously, there was no such reduction in efficacy, but a continuous improvement. The efficacy of P6_CpNPP_ lineages approached the level of CpGV-R5 on this host (LC_50_ for CpGV-R5 was 22.43 (13.73–34.36); LC_50_s for P6_CpNPP_ ranged between 28.24 and 40.77). This trend was independent of the original frequency of the parent genotypes. The hypothesis of progressive replacement of one genotype by the second in the experimental populations could explain the observed behavior. Over passages, CpGV-M would be expected to reduce progressively and disappear. PCR analysis of the virus lineages revealed that virus lineages replicating in a fully-permissive host retained the markers of both genotypes, while improving their ability to control the resistant insects; conversely, in virus lineages submitted to strong selection, the CpGV-M-specific marker was absent or highly reduced, and the efficacy was not as good (Figure 4).

It would also be expected that the Pi_RGV_ populations, submitted to a higher selective pressure for CpGV-R5 genotypes, would recover the efficacy of this isolate quicker than Pi_CpNPP_. However, we observed that when replication was carried out on R_GV_ insects (populations Pi_RGV_), the LC_50_s did not reach the level of the P0_RGV_ virus populations, neither on CpNPP nor on R_GV_, after six passages. Increasing the number of passages and reducing the multiplicity of infection would eventually lead to a highly-pathogenic isolate similar to CpGV-R5, following a process similar to that used to select this isolate.

According to our results, allowing the virus isolates to adapt to the specific host populations resulted in similar levels of control, while preserving the virus genetic diversity.

In CM populations in orchards, there was a variable proportion of resistant individuals, mixed with susceptible individuals. This was reflected by a two-plateau dose mortality line when challenged with a virus.

Using a single virus genotype against host populations with variable genetic background can result in failures of the control due to the variable susceptibility of the host to that virus genotype. Mixed virus populations will be able to colonize each individual of the host population. Providing genetic diversity is supposed to facilitate adaptation of populations to a new environment. For a virus, this new environment can be a host harboring a new resistance trait or the presence in the host of an inhibition molecule.

In apple orchards, the success of control of the codling moth with CpGV requires massive release of the virus. The usual dose in the field is between 3 × 10^12^ and 10^13^ OB/Ha [39,40]. Glen and Payne [41] applied 9 × 10^13^ OB/Ha and calculated 2404 ± 1608 OBs/mm^2^ of leaves or fruit. Accordingly, the estimate concentration obtained with commercial products would be at least 271 OB/mm^2^ in the orchards. Taking into account the observations on the size of the feeding holes excavated by neonate larvae [42] and assuming a homogeneous dispersion of the OBs, a neonate larvae would ingest an average of 2.76 OB after only 3.5 min of contact with the leaves or the fruit, raising quickly to a hundred OBs in less than 1 h. Accordingly, in field conditions, larvae likely ingest multiple OBs.

A recent analysis of the commercial virus preparations developed to fight field resistance showed that they were genotypically diverse, contrary to the products commercialized before 2005, when pure CpGV-M was the active ingredient. The two products analyzed contained at least three genotypes each [43], among them CpGV-M and a type I resistance breaking genotype, specific for each product. In these conditions, larvae likely ingest more than a single genotype. However, no information about the outcome of infections in the field is available, due to the early death of the larva and the difficulty of monitoring in the field. Simulating field infection in the laboratory would shed light on the frequency of multiple infections in CM.

## Figures and Tables

**Figure 1 viruses-11-00621-f001:**
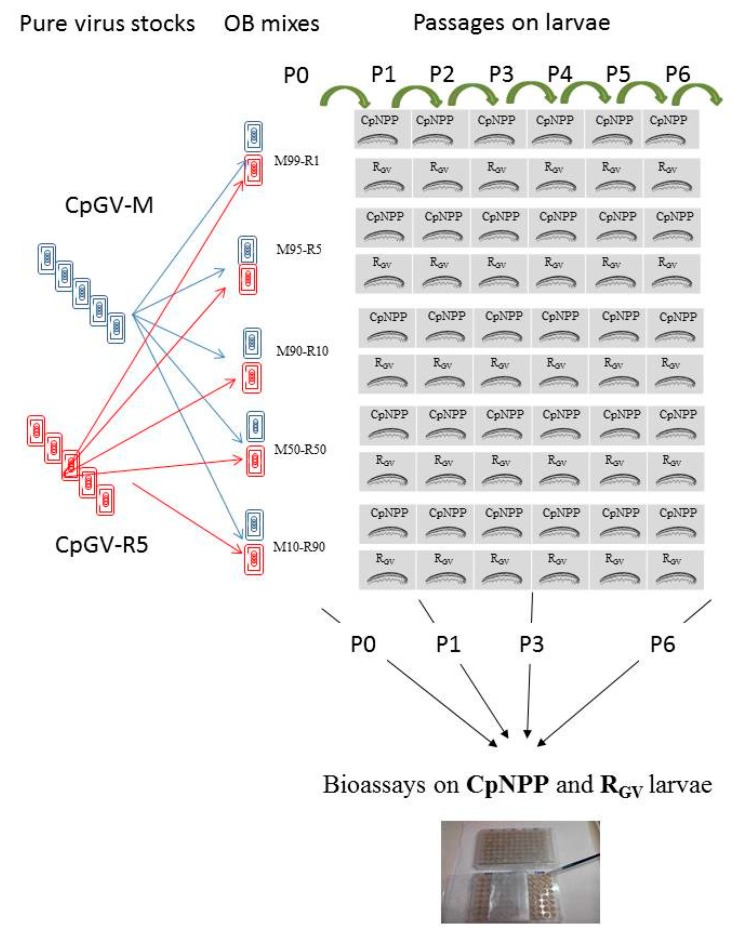
Schematic representation of the different mixed genotype lineages obtained by passaging mixed genotype virus populations on susceptible (CpNPP) and resistant (R_GV_) insects. The bioassays on CpNPP and R_GV_ neonate larvae were performed using Passage 1 (P1), P3, and P6 viruses. P0 (previously published) were used as a reference. OB, occlusion body.

**Figure 2 viruses-11-00621-f002:**
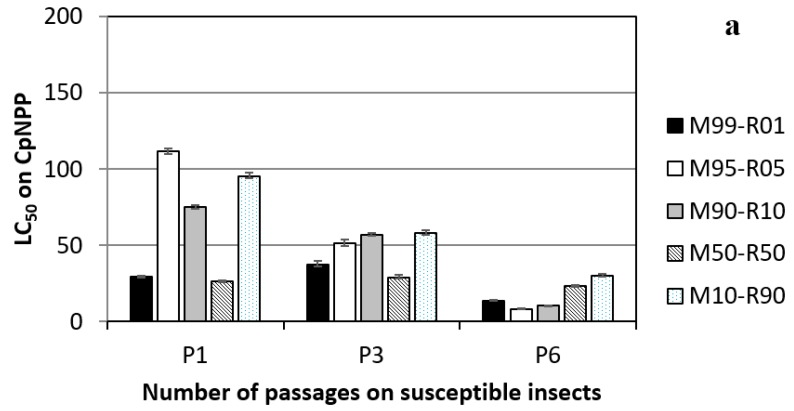
Efficiency (LC_50_) of the different experimental virus populations replicated on CpNPP and tested on (**a**) CpNPP and (**b**) R_GV_. Parental CpGV-M and CpGV-R5 alone had LC_50_s on CpNPP and R_GV_ of 13.10 OB/µL and 6.76 OB/µL, respectively.

**Figure 3 viruses-11-00621-f003:**
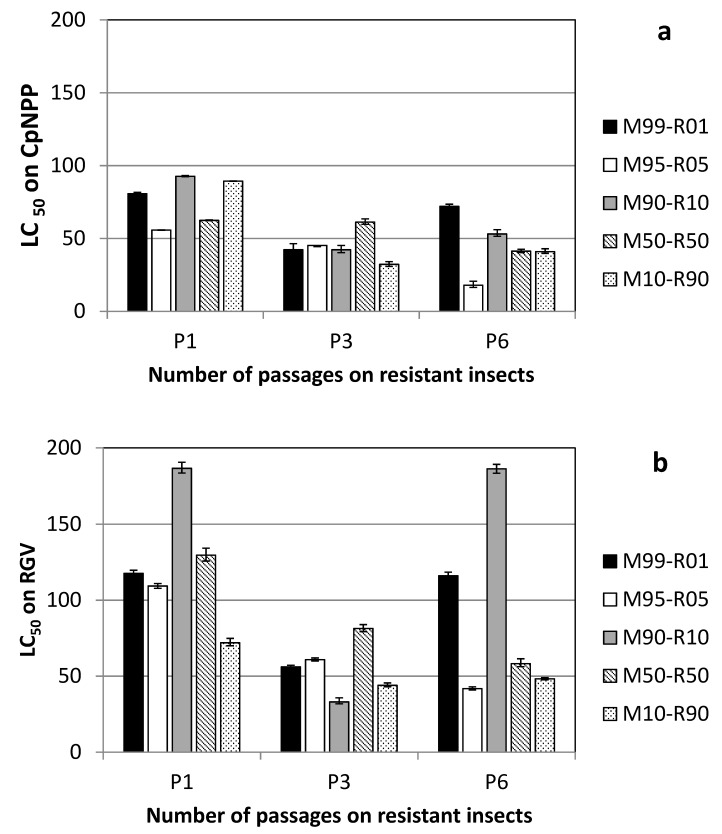
Efficiency (LC_50_) of the different experimental virus populations replicated on R_GV_ and tested on (**a**) CpNPP and (**b**) R_GV_. Parental CpGV-M and CpGV-R5 alone had LC_50_s on R_GV_ of 2.22 × 10^6^ OB/µL and 22.43 OB/µL, respectively.

**Figure 4 viruses-11-00621-f004:**
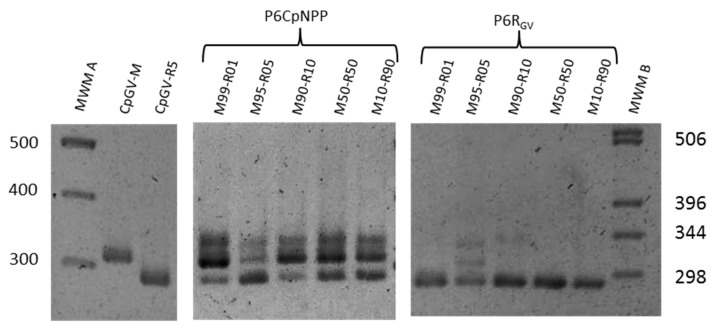
Gel electrophoresis of PCR products revealing the variability on the *pe38* region of *Cydia pomonella* granulovirus populations successively produced during six generations on susceptible (P6_CpNPP_) and resistant (P6_RGV_) insects. CpGV-M and CpGV-R5 are presented as references. MWM A: Molecular weight marker GeneRuler 100-bp DNA Ladder (Fermentas, Burlington, ON, Canada). MWM B: 1-kb DNA Ladder (Invitrogen, Carlsbad, CA, USA).

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
