# Peer review of "Importance of the Host Phenotype on the Preservation of the Genetic Diversity in Codling Moth Granulovirus"

_viruses, 2019, doi:10.3390/v11070621_

Round 1

Reviewer 1 Report

Graillot et al. examine the propagation of mixtures of two different granulovirus isolates, CpGV-M (M) and CpGV-R5 (R), in two different codling moth colonies, CpNPP and Rgv, during six passages. Since Rgv is resistant to M, the monitoring of differences in pathogenicity and composition is of scientific and economic interest. The general conclusion that in CpNPP the diversity is conserved while on Rgv the proportion of M is drastically reduced in the course of the experiment is supported by the data.

However, major parts of the manuscript are not conclusive. The authors state that they use the original mixes (P0) as a reference. However; the LC50 of the pure isolates is hidden in the Methods part and legends of Fig. 2 and 3, the LC50 of the M90R05 and M50R50 mixture has to be looked up in ref. 26 and the LC50 of ther other P0 mixtures are still unknown to me. Without that information, the Bioassay results cannot be interpreted.

According to ref. 26, the LC50 of the original mixtures on CpNPP should be in the range of 6 to 13 OBs/ul. P1 suddenly shows LC50s of over 80 OBs/ul, even after passage in CpNPP. Those LC50s are, however, not consistent and apparently also not related to the initial composition of P0. All Bioassay data in Fig. 2 and 3 are not conclusive, e.g. the LC50 of M90R10 after passage in Rg shows the highest LC50 in P1 (comparable to P0), the lowest in P3 and again the highest in P6 when testing on Rg. These discrepancies are not explained. The authors speculate in the discussion that recombination might lead to some less adapted gene combinations in P1 and this effect does not appear in M99R01 due to the high frequency of infetions with only one genotype. This rise should then also be observed in M50R50 after passaging in CpNPP when testing on CpNPP, where is is missing. On the other hand, the LC50 of the same P1 on Rgv is considerably higher than the corresponding P0. Taken together, the LC50 data is very contradictory.

In the introduction, the authors mention that for a mixed infection, an ingestion of more than one OB is nesessary but leave the question open how many OBs are estimated to be ingested per larva. In the methods part, we learn that the concentration in the medium is 40OB/ul (which would mean many ingested OBs), and the discussion stated that in the field, larve ingest a hundred OBs in less than an hour and "we have chosen conditions in which each larva would eat more than one OB". Does "more than one" mean 2, 200 or 2000? The authors speculate that the lack of rise in LC50 in M99R01 is due to the high frequency of infections with only one genotype. The authors should come up with an estimate how many OBs are ingested per larva so that the reader can judge the probability of multiple infections.

The authors state in the methods part that pure M and pure R populations were used as "control populations" (p.3 l.111-112). Since there are no results of these populations, the authors probably men "reference population".

How many of the 48 larvae used were yielded ("presenting signs of viral infection") in each passage? According to the Methods section, the concentration used was 40 OBs/ul of diet. For the M90R10 mixture, ref. 26 states an LC50 of about 200 OBs/ul on Rgv. The LC50 for M99R01 is probably even higher. Why do the authors state in the Methods section (p.3l.120) that a mortality of 90% is expected? E.g., of how many laeva was the P1 of M99R01 on Rgv extracted?

Fig. 2 and 3 should be scaled to the same Y axis to be comparable.

While the authors explicitely mention the genetic diversity of the virus in the abstract, the composition of the offspring viruses was just examined by gel electrophoresis (Fig. 4), but not quantified. These results are not consistent - while, after six passages in CpNPP, the M95R05 mixture shows mainly R offspring, all other mixtures show more M offspring. For a better interpretation of Fig. 4, the authors should include PCRs of some P0 mixtures as a reference so the ratio between the two isoforms can be estimated from the electrophoresis pictures. Can the ratio be quantified approximately? Does the M isolate only disappear in P6 after passaging in Rgv or also in the earlier passages? The time course of the M/R ratio in the different passages would be interesting, especially the change of the M fraction during Rgv passages. The starting material for the PCR in Fig. 4 should only be OBs that contain nothing besides viruses. Did the authors examine the third band which appears in the CpNPP passages? Where does this band start to appear?

The authors state in the Methods section (p. 3 l. 142) that the pe38 fragment is 295bp for M and 315bp for R. However, in Fig. 4 the weights seem to be the other way round (R smaller than M).

In the Results section (p. 5 l. 175-176), the authors state that a passage on Rgv would mean a "strong selection to the A genotype. However, in the introduction (p.2, l.68-69), they state that M belongs to group A and R belongs to group E. Thus, the selection pressure would be towards group E.

In Fig. 1, the meaning of the black and gray virus pictograms on the left is not clear. What should they tell the reader?

Minor comments:
p. 2, line 55: "Restriction Fragment Length Polymorphism" instead of "Restriction Enzyme Length Polymorphism"
p. 2 line 67: "Analyzed in more detail" instead of "Analyzed more in detail"
p. 3 line 142: "bp" instead of "pb"
p. 4 line 166; p. 5 line 178, p. 8 l. 251: "fraction", "content", "ratio" or "composition" instead of "frequency"
p. 5, line 176: "group" instead of "groupe"
p. 8 l. 251: "trend" instead of "evolution"
Acknowledgements are still the dummy text.
Fig. 2 "LC50" instead of "CL50".

Author Response

Dear reviewer,

Thanks for you careful analysis of our manuscript. Your review helped us to improve it; We hope that this sercond version will satisfy your criticisms. Enclosed a pdf file with a point by point reply

Reviewer 2 Report

The manuscript describes the investigation of phenotypic and genotypic diversity in mixed Cydia pomonella granuloviruses (CpGVs)in vivo. CpGVs isolated after passages in virus-susceptible insects were more infectious than those isolated after passages in virus-resistant insects and p38 gene in CpGVs isolated after passages in virus-susceptible insects showed the genetic diversity. These results are interesting for the control of insect pests. This manuscript is well-documented, but still has some doubts and points to be improved. This manuscript should be revised before its acceptance to this journal “Viruses”. Some comments are described below.

1. The authors did PCR to amplify some regions of pe38 gene. The authors should describe the sequences of primers for this PCR even though the paper referred as [26] shows these sequences

2. The Figure 1 should be revised to be made more clear.

3. In Figure 4, the additional band over 315 bp was detected in each p6CpNPP sample. Why did this band appear? Was DNA sequencing of this band carried out? It would be possible to deduce how to generate this band by its sequencing.

4. The authors should mention the function of pe38 gene product in brief.

5. The discussion should be summarized more because it is lengthy.

Author Response

Dear reviewer,

Thanks for you careful analysis of our manuscript. Your review helped us to improve it; We hope that this second version will satisfy your criticisms. Enclosed a pdf file with a point by point reply

Reviewer 3 Report

In this manuscript, the author tried to describe the stable in genetic diversity after 6 passages of 2 different CpGV strain (M and R5) to the codling moth colony (CpNPP) as well as RGV. At first, what is the importance of this study? It should be clarified to the reader. For example, the increase of genetic diversity of CpGV is easy to happen during the passage, and this genetic change will decrease the host susceptibility? Another concern is about the molecular marker for detecting the genetic changes during 6-passages. how does the author confirm it is true that the pe38 could represent this event? The genetic change may also happen in other location of the CpGV genome (the hr region or some gene involved in host range determination, i.e. iap gene)    

-          Why important of viral genetic diversity change? The author should explain it in the manuscript, though in the Line 72-86, there are some explanations. Does the genetic variation of CpGV will decrease the virulence against the host? 

-          In the abstract, the author mentioned “markers characteristic”, please detail explain it?

-          In figure 2, what is CL50 besides, there should have data of 6 passages (P1-P6)?

-          Line170, LC50, and LC90? But there is no data of LC90

-          As the materials and Method section described that the R5 strain has higher virulence than M strain. How to explain the LC50 data of M95-R05, M90-R10 and M10-R90 on CpNPP in P1?

-          Line187 How many replications of the trails in this study? If there are statistically repeats, the molecular data should have repeat; if so, I suggest transform the data to semi-quantity data. It would be easy to read

Author Response

Dear reviewer,
Thanks for you careful analysis of our manuscript. Your review helped us to improve it; We hope that this second version will satisfy your criticisms. Enclosed a pdf file with a point by point reply
In this manuscript, the author tried to describe the stable in genetic diversity after 6 passages of 2 different CpGV strain (M and R5) to the codling moth colony (CpNPP) as well as RGV. At first, what is the importance of this study? It should be clarified to the reader. For example, the increase of genetic diversity of CpGV is easy to happen during the passage, and this genetic change will decrease the host susceptibility?

Response: We could expect that genetic diversity would be maintained upon passage, but this is not what we observe when replicating in RGV host. On the other hand, we would expect that mixed virus populations would progressively adapt to the host in which they replicated, losing adaptation to the other host (so, increasing the virus doses required for control). Again, this does not happen. We think that these observations challenge the way we are using this viruses to control insects.

Another concern is about the molecular marker for detecting the genetic changes during 6-passages. how does the author confirm it is true that the pe38 could represent this event? The genetic change may also happen in other location of the CpGV genome (the hr region or some gene involved in host range determination, i.e. iap gene)

Response: The ability for a CpGV virus to replicate in a type I resistant host has been associated to the presence of this marker at the level of the pe38 gene. We have added more information on the introduction to make this point clearer (Line 75): “A modification of the viral pe38 gene, whose function remain unknown, has been associated to the ability of a virus to replicate in a type I resistant host, like RGV (ref). CpGV-M and CpGV-R5 differ by a small 24 bp duplication present in CpGV-M, but absent in CpGV-R5. In type I resistant hosts, CpGV-M replication is blocked at an early stage in all cells of the larva.”

Why important of viral genetic diversity change? The author should explain it in the manuscript, though in the Line 72-86, there are some explanations. Does the genetic variation of CpGV will decrease the virulence against the host?

Response: We have added a sentence in the introduction on the increased pathogenicity of virus populations composed of various genotypes on SfNPV, and on CpGV (Line 86): “In NPVs, multiple genotypes can be occluded in the same OB [24], one OB can thus represent a sample of the population diversity, and consequently a larva that eats a single OB can be infected by a variety of virus genotypes. In GVs however, genotypic diversity in a single larva would rely on the larva eating more than one OB. In a previous study, we have investigated the ability of mixes of CpGV isolates to control a resistant colony. We observe that mixtures of OBs are more efficient than expected on the control of the hosts larvae, be they susceptible or resistant [30].”

In the abstract, the author mentioned “markers characteristic”, please detail explain it?

Response: We have added the following sentence to the abstract: “CpGV-M and CpGV-R5 differ by their ability to replicate in codling moth larvae carrying the type I resistance. This ability is associated to a genetic marker located in the virus pe38 gene.” and “host colony (RGV) that carries the type I resistance, and thus blocks CpGV-M replication.”

In figure 2, what is CL50 besides, there should have data of 6 passages (P1-P6)?

Response: We have represented only the data of three passages, as the figure with the 6 passages x5 lineages would be unreadable. We chose to present one at the beginning, one in the middle and the last passage.

Line170, LC50, and LC90? But there is no data of LC90

Response: Sorry about that, it was an error, only the data of LC50 are presented on the Figure.

As the materials and Method section described that the R5 strain has higher virulence than M strain. How to explain the LC50 data of M95-R05, M90-R10 and M10-R90 on CpNPP in P1?

Response: The difference between CpGV-M and CpGV-R5 on fully permissive hosts (CpNPP) is not important (13 to 7 OBS/μL). On resistant hosts (RGV), this difference is extremely high (2.2*10E6 to 22 OB/μL).

In previous work we analyzed the pathogenicity of pure virus genotypes (CpGV-M and CpGV-R5) and mixtures of these genotypes in various proportions. We observed that mixtures that contain low proportions of CpGV-R5 were able to kill RGV larvae, while pure CpGV-M did not. We conclude that the presence of CpGV-R5 virus allows replication of CpGV-M. In addition, the efficacy of these mixses was higher that the efficacy of the same amount of CpGV-R5 alone, thus, CpGV-M contributes to the efficacy.

The OBs produced when larvae are fed with mixtures of pure genotype OBs (that is, PI), are less pathogenic than the original mix. We comment this point more in detail following the request of reviewer 1. However, when replicating in fully permissive hosts, these lineages increase pathogenicity over passages. When replicating on resistant hosts, this is not he case.

Line187 How many replications of the trails in this study? If there are statistically repeats, the molecular data should have repeat; if so, I suggest transform the data to semi-quantity data. It would be easy to read.

Response: There as a single replicate of each lineage, starting from 48 L3 larvae, and three replicated of each bioassay, that is, 10 lineages and 180 independent bioassays.

Round 2

Reviewer 1 Report

All points have been addressed by the authors.

In line 76, it should read "whose function remainS unknown".

Reviewer 3 Report

The authors have addressed my comments, therefore I suggest accept this manuscript as current version.